# Stretchable and Conductive Cellulose/Conductive Polymer Composite Films for On-Skin Strain Sensors

**DOI:** 10.3390/ma15145009

**Published:** 2022-07-19

**Authors:** Joo Won Han, Jihyun Park, Jung Ha Kim, Siti Aisyah Nurmaulia Entifar, Ajeng Prameswati, Anky Fitrian Wibowo, Soyeon Kim, Dong Chan Lim, Jonghee Lee, Myoung-Woon Moon, Min-Seok Kim, Yong Hyun Kim

**Affiliations:** 1Industry-University Cooperation Foundation, Pukyong National University, Busan 48513, Korea; hanjoo1020@naver.com; 2Department of Smart Green Technology Engineering, Pukyong National University, Busan 48513, Korea; ds3dem@naver.com (J.P.); lagamuffin@naver.com (J.H.K.); nurmauliaentifar29@gmail.com (S.A.N.E.); ajengprameswati97@gmail.com (A.P.); ankiz118248@gmail.com (A.F.W.); 3Surface Technology Division, Korea Institute of Materials Science (KIMS), Changwon 51508, Korea; kimso1965@kims.re.kr (S.K.); dclim@kims.re.kr (D.C.L.); 4Department of Creative Convergence Engineering, Hanbat National University, Daejeon 34158, Korea; jonghee.lee@hanbat.ac.kr; 5Department of Materials and Life Science Research Division, Korea Institute of Science and Technology, Seoul 02792, Korea; mwmoon@kist.re.kr (M.-W.M.); nanostructures@kist.re.kr (M.-S.K.); 6School of Electrical Engineering, Pukyong National University, Busan 48513, Korea

**Keywords:** on-skin sensors, stretchable, cellulose, PEDOT:PSS, wearable electronics

## Abstract

Conductive composite materials have attracted considerable interest of researchers for application in stretchable sensors for wearable health monitoring. In this study, highly stretchable and conductive composite films based on carboxymethyl cellulose (CMC)-poly (3,4-ethylenedioxythiopehe):poly (styrenesulfonate) (PEDOT:PSS) (CMC-PEDOT:PSS) were fabricated. The composite films achieved excellent electrical and mechanical properties by optimizing the lab-synthesized PEDOT:PSS, dimethyl sulfoxide, and glycerol content in the CMC matrix. The optimized composite film exhibited a small increase of only 1.25-fold in relative resistance under 100% strain. The CMC-PEDOT:PSS composite film exhibited outstanding mechanical properties under cyclic tape attachment/detachment, bending, and stretching/releasing tests. The small changes in the relative resistance of the films under mechanical deformation indicated excellent electrical contacts between the conductive PEDOT:PSS in the CMC matrix, and strong bonding strength between CMC and PEDOT:PSS. We fabricated highly stretchable and conformable on-skin sensors based on conductive and stretchable CMC-PEDOT:PSS composite films, which can sensitively monitor subtle bio-signals and human motions such as respiratory humidity, drinking water, speaking, skin touching, skin wrinkling, and finger bending. Because of the outstanding electrical properties of the films, the on-skin sensors can operate with a low power consumption of only a few microwatts. Our approach paves the way for the realization of low-power-consumption stretchable electronics using highly stretchable CMC-PEDOT:PSS composite films.

## 1. Introduction

Conductive materials have attracted increasing attention in stretchable sensors owing to their high conductivity, cost-effectiveness, excellent mechanical properties, and ease of processability [1,2]. Even though on-skin sensors based on highly conductive materials show small resistance changes in strain sensors for small input signals compared to low-conductivity materials, they have exceptional merits in terms of low power consumption, fast response, and low hysteresis [3,4,5]. Over the last several decades, conductive polymers, metal oxides, metal nanowires, and carbon-based materials have been investigated as conductive sensing materials for high-performance on-skin strain sensors [6,7,8,9]. The conductive polymer poly (3,4-ethylenedioxythiophene):poly (styrenesulfonate) (PEDOT:PSS) is a promising material with high conductivity, solution processability, low cost, and mechanical robustness [10,11,12]. However, conventional composite films based on PEDOT:PSS and polymeric matrices have low stretchability and poor electrical conductivity. Wang et al. reported a polyvinyl alcohol/carboxymethyl cellulose/PEDOT:PSS hydrogel with a conductivity of 75 S/cm for electrocardiogram monitoring [13]. Gao et al. developed a conductive PEDOT:PSS/polyvinyl alcohol composite fiber with an electrical conductivity of 3.6 ± 0.1 S/cm for wearable sensors [14]. Sun et al. fabricated a PEDOT:PSS/polyacrylamide hydrogel for flexible strain sensors, which has an electrical conductivity of 6.0 × 10^−4^ S/cm [15].

Cellulose is regarded as a promising matrix material for stretchable conductive composites owing to its environmental friendliness, biocompatibility, biodegradability, nontoxicity, natural abundance, sustainability, and superior mechanical properties [16,17]. The massive number of hydroxyl and carboxyl functional groups of cellulose easily form strong bonds with polymer materials, enabling high-performance composites [18]. Cellulose-based composites can be used for various electronic applications.

In this study, we developed a highly stretchable on-skin strain sensor based on a conductive carboxymethyl cellulose (CMC)/lab-synthesized PEDOT:PSS composite (CMC-PEDOT:PSS) film. The CMC-PEDOT:PSS composite film exhibited high electrical and mechanical robustness under mechanical deformation. The multifunctional sensors based on the conductive CMC-PEDOT:PSS film successfully monitored various bio-signals with high sensitivity, skin conformability, and power consumption, as low as a few microwatts. The results suggest that the stretchable and conductive CMC-PEDOT:PSS film has great potential for wearable health monitoring and can be applied to a variety of stretchable electronics.

## 2. Materials and Methods

### 2.1. Synthesis of PEDOT:PSS Solution

Polystyrene sulfonate (PSS) dissolved in DI water (4.0 wt%), sodium persulfate (Na_2_S_2_O_8_), iron (III) sulfate (Fe_2_(SO_4_)_3_), and DI water (400 g) were mixed and stirred for 1 h at room temperature in argon atmosphere. To further increase the conductivity and stretchability, 1 wt% of CSE100 (AH Materials, Busan, Korea) was introduced into the solution as a conductivity/stretchability enhancer. The 3,4-ethylenedioxythiophene (EDOT) monomer was then added to the solution. The mixture solution was stirred continuously for 20 h at room temperature. The molar ratio of EDOT:Na_2_S_2_O_8_ and EDOT:Fe_2_(SO_4_)_3_ was 1:0.9 and 1:0.02, respectively. A dark blue color of the PEDOT:PSS solution was produced, and the PEDOT:PSS weight ratio was fixed at 1:2.

### 2.2. Preparation of Stretchable CMC-PEDOT:PSS Composite Film

Carboxymethylcellulose (CMC) sodium salt (Mw ~250,000), dimethyl sulfoxide (DMSO), and glycerol were purchased from Sigma-Aldrich. The stretchable CMC-PEDOT:PSS composite film was prepared as follows. First, 0.2 g of CMC was diluted with 10 g of deionized water, and then stirred at 40 °C. Subsequently, PEDOT:PSS (2.2, 2.6, and 3.0 g), DMSO (1.4, 1.8, 2.2, and 2.6 g), and glycerol (0.2, 0.4, 0.6, 0.8, and 1.0 g) were added to the CMC solution and stirred at 40 °C until well dispersed. The prepared solutions were poured over a petri dish, and then dried for a day in an oven at 50 °C. Stretchable cellulose films were obtained by peeling them from the petri dishes.

### 2.3. Characterization of Stretchable CMC-PEDOT:PSS Composite Film

The sheet resistance was measured using the van der Pauw method with a source measurement unit (Keithley 2401). The surface morphologies of the films were recorded by field-emission scanning electron microscopy (SEM; MIRA3, TESCAN, Brno, Czech Republic). Attenuated total reflectance-Fourier transform infrared (ATR-FTIR) spectroscopy was performed using a Fourier-transform infrared spectrometer (CARY 600, Bruker, Karlsruhe, Germany). The tensile test was examined using a Keithley source measure unit on a custom-made stage. Both ends of the CMC-PEDOT:PSS composite films were wired to a source measure unit for characterizing sensing performance.

## 3. Results and Discussion

A highly stretchable and conductive CMC-PEDOT:PSS composite film was fabricated using the process shown in Figure 1a. First, CMC, PEDOT:PSS, DMSO, and glycerol were blended until they were well dispersed. The chemical structures of the materials are shown in Appendix A. Next, the mixture was poured onto a Petri dish and dried in an oven at 50 °C for one day. Finally, a stretchable and conductive CMC-PEDOT:PSS film was obtained by peeling it off from the Petri dish. The prepared CMC-PEDOT:PSS film adhered conformably to human skin owing to its excellent conformability and elasticity (Figure 1b). Figure 1c shows the photographs of the CMC-PEDOT:PSS films under the tensile strains of 0% and 100%. The film could be easily stretched to 100% owing to its excellent elasticity.

The CMC-PEDOT:PSS film was optimized by controlling the amount of PEDOT:PSS, DMSO, and glycerol in the CMC matrix. The changes in the relative resistance of the film with respect to the applied strain are shown in Figure 2a–c. The corresponding sheet resistances are shown in Appendix A. As shown in Figure 2a, the relative resistance of the composite films (CMC:DMSO = 1:11 and CMC:glycerol = 1:5) increased by 2.04-, 1.13-, and 3.14-fold with increasing CMC to PEDOT:PSS volume ratios of 1:11, 1:13, and 1:15, respectively, when stretched to 100% strain. An optimum amount of PEDOT:PSS was observed for the film with a ratio of 1:13 (Figure 2a). The thicknesses of the films with the CMC to PEDOT:PSS ratios of 1:11, 1:13, and 1:15 were 41, 37, and 34 μm, respectively. It was observed that the thickness of film did not greatly affect the strain-induced electrical properties. The relative resistance of the films (CMC:PEDOT:PSS = 1:13 and CMC:glycerol = 1:5) increased from 1.26 to 1.82-fold as the CMC to DMSO ratio increased from 1:7 to 1:13 at 100% strain (Figure 2b). The relative resistance of the films (CMC:PEDOT:PSS = 1:13 and CMC:DMSO = 1:9) with the CMC to glycerol ratios of 1:1, 1:2, 1:3, 1:4, and 1:5 were 1.93, 1.25, 2.20, 1.85, and 1.82, respectively, when stretched to 100% strain (Figure 2c). The results showed that the optimum ratios of CMC to PEDOT:PSS, CMC to DMSO, and CMC to glycerol were 1:13, 1:9, and 1:2, respectively. The optimum film (conductivity: ~1.0 × 10^−2^ S/cm) exhibited a limited resistance, where the sheet resistance of the optimum film increased from 5500 ohm/sq to 6890 ohm/sq (1.25-fold increase in resistance) when stretched to 100% strain. The sheet resistances of films fabricated under various conditions are shown in Appendix A. Literature data for cellulose-PEDOT-based composites is listed in Appendix A.

The mechanical robustness of the optimized CMC-PEDOT:PSS composite film was investigated by conducting tape attachment/detachment, cyclic bending, and cyclic stretching/releasing tests (Figure 2d–f). The relative resistance of the composite film changed slightly during tape attachment/detachment tests, which increased only by a factor of 1.13 after 20 cycles (Figure 2d). The film also exhibited a limited resistance change under cyclic bending tests after 500 bending cycles at a bending radius of 5 mm (Figure 2e). Under the cyclic stretching/releasing test at a strain of 50%, the film exhibited a slight increase in relative resistance (1.65-fold) after 100 cycles of stretching. The outstanding mechanical robustness of the CMC-PEDOT:PSS composite film is attributed to the excellent electrical contact between the PEDOT:PSS domains and the strong bonding strength between the CMC and PEDOT:PSS molecules, which effectively prevents mechanical deformation of the films.

Figure 3a shows the SEM images of CMC and CMC-PEDOT:PSS films under the strains of 0% and 100%. The CMC film exhibited a smooth surface at a strain of 0%. The surface of CMC film was remarkably deformed, with a large number of cracks and wrinkles, under an applied strain of 100%. The CMC-PEDOT:PSS composite film displayed a rough and tangled surface at 0% strain. Despite the high strain applied (100%), the composite film maintained its initial surface state without noticeable cracks or morphological changes. The CMC-PEDOT:PSS composite film without structural changes, even at high strain, exhibited very stable electrical properties under mechanical deformation. Figure 3b shows the FTIR spectra of the CMC and CMC-PEDOT:PSS films. The broad band at 3000–3700 cm^−1^ corresponds to −OH groups, and the peaks at 2800–3000 cm^−1^ are attributed to the C−H stretching vibration of the CMC film. The peak at 1594 cm^−1^ corresponds to COO− groups. The CMC-PEDOT:PSS composite film exhibited decreased intensities of the peaks (−OH, C−H, and COO− groups) compared to the CMC film. Furthermore, noticeable peak changes in the range of 880 to 1150 cm^−1^ were observed owing to the introduction of PEDOT:PSS, where the peaks at 950 and 1012 cm^−1^ correspond to C−S bonds in the thiophene rings and s-phenyl bonds in the sulfonic acid of PEDOT:PSS, respectively.

We demonstrate multifunctional sensors based on optimized CMC-PEDOT:PSS composite films to monitor subtle biosignals from the human body, such as respiratory humidity, drinking water, speaking, and various human movements (Figure 4). In Figure 4a, the humidity sensor was attached inside the mask, which detected respiratory humidity and responds to changes in resistance during exhalation and inhalation. The relative resistance of the humidity sensor sensitively decreased and increased during the exhale and inhale processes, respectively. During the exhale process, water adsorption leads to dissociation of the hydroxyl groups (–OH) of PSS and increases the negative charges in PSS. The increase in negative charges in PSS causes an effective Coulombic attraction between the positively charged PEDOT and the negative charges. This process decreases the molecular distance between PEDOT and PSS, and the resistance of PEDOT:PSS. The composite film conformally attached to the throat responded sensitively to the motion of throat muscle in response to drinking water, showing a decrease of approximately 40% in relative resistance change during drinking (Figure 4b). Furthermore, the on-skin sensors on the throat were able to monitor subtle vibrations of the larynx in real time via phonation such as “Hello,” “Cellulose,” “Conducting,” and “Polymer” (Figure 4c–f). The on-skin sensors can perceive the accent of words with a change in resistance. The resistance change decreased slightly when speaking with a weak accent (<10%), whereas it decreased significantly to approximately 20–40% while speaking with a strong accent. It is important to note that the sensors based on the conductive CMC-PEDOT:PSS composites exhibited very low power consumption of only a few microwatts (0.9–7.6 μW at 0.2 V). Moreover, the on-skin sensors successfully detected various human motions such as skin touching, skin wrinkling, and finger bending (Figure 4g–i). The relative resistance change of the on-skin sensors decreased by approximately 20% in a response to a finger touch and recovered the original resistance after taking the finger off (Figure 4g). When the on-skin sensor attached to the back of the hand was repeatedly wrinkled and released, the relative resistance change of the sensor decreased by approximately 20% (Figure 4h). The relative resistance of the sensor attached to the finger increased by approximately 30% when the finger was bent. The resistance almost returned to its initial state when the finger was straightened. Sensors with conductive CMC-PEDOT:PSS composite films sensitively monitor subtle bio-signals and human movements with very low power consumption and high sensitivity.

## 4. Conclusions

We present highly stretchable and conductive CMC-PEDOT:PSS composite films. The electrical and mechanical properties of the films were carefully optimized by controlling the lab-synthesized PEDOT:PSS, DMSO, and glycerol content in the CMC matrix. The relative resistance of the optimized composite film increased only 1.25-fold under 100% strain. The CMC-PEDOT:PSS composite film exhibited a limited resistance change under cyclic tape attachment/detachment and bending tests, indicating superior mechanical robustness. These results suggest that the strongly linked CMC and PEDOT:PSS molecules efficiently provide high conductivity and prevent the mechanical deformation of the films. The stretchable and conductive CMC-PEDOT:PSS composite films were successfully applied to multifunctional sensors to detect subtle bio-signals and human movements, such as respiratory humidity, drinking water, speaking, skin touching, skin wrinkling, and finger bending in real time. High-performance on-skin strain sensors show a low power consumption of only a few microwatts owing to the high conductivity of the films. We believe that the highly stretchable and conductive CMC-PEDOT:PSS composite holds great promise for next-generation, low-power consumption wearable electronics.

## Figures and Tables

**Figure 1 materials-15-05009-f001:**
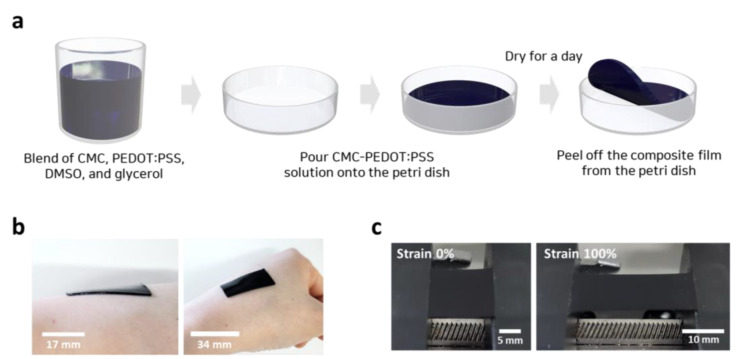
(**a**) Schematic of fabrication process for CMC-PEDOT:PSS composite films. (**b**) Photographs of CMC-PEDOT:PSS film attached to human skin. (**c**) Photographs of CMC-PEDOT:PSS films under strains of 0% and 100%.

**Figure 2 materials-15-05009-f002:**
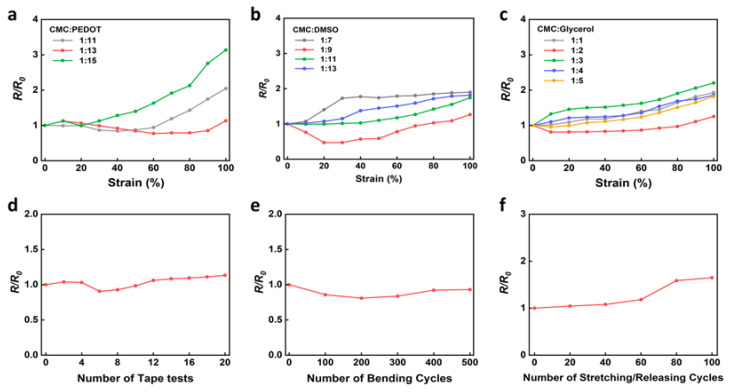
Relative resistance of CMC-PEDOT:PSS films as a function of ratios of CMC to (**a**) PEDOT:PSS, (**b**) DMSO, and (**c**) glycerol. Relative resistance of the optimized CMC-PEDOT:PSS composite film (CMC:PEDOT:PSS = 1:13, CMC:DMSO = 1:9, and CMC:glycerol = 1:2) under (**d**) tape attach/detach, (**e**) cyclic bending, and (**f**) cyclic stretching/releasing tests.

**Figure 3 materials-15-05009-f003:**
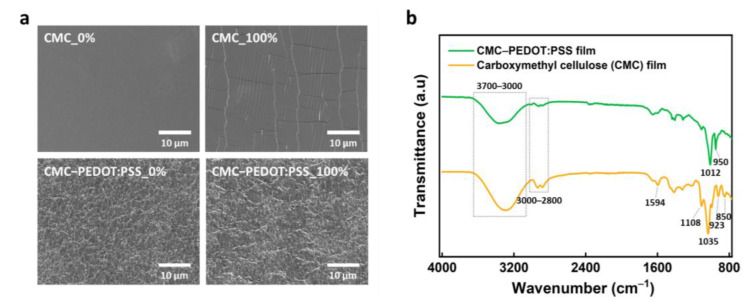
(**a**) SEM images of CMC and CMC-PEDOT:PSS films under strain of 0% and 100%. (**b**) FTIR spectra of CMC and CMC-PEDOT:PSS films.

**Figure 4 materials-15-05009-f004:**
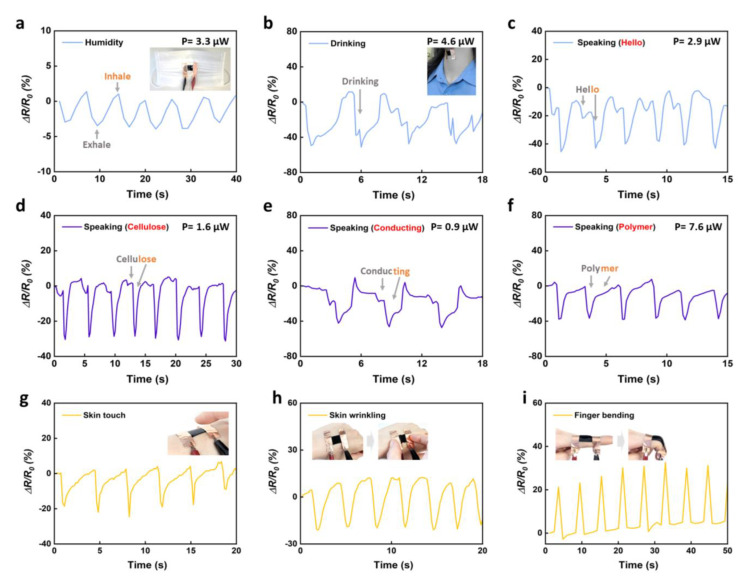
Response curves of on-skin sensors based on CMC-PEDOT:PSS film for (**a**) respiratory humidity, (**b**) drinking, and speaking the words (**c**) hello, (**d**) cellulose, (**e**) conducting, and (**f**) polymer. Changes in relative resistances of sensors monitoring human movements of (**g**) skin touch, (**h**) skin wrinkling, and (**i**) finger bending.

## Data Availability

Not applicable.

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
