# Peer review of "Stretchable and Conductive Cellulose/Conductive Polymer Composite Films for On-Skin Strain Sensors"

_materials, 2022, doi:10.3390/ma15145009_

Round 1
Reviewer 1 Report
1. Authors have to provide clear figures.
2. It will be good to provide a curve between resistance and strain for any samples.
3. There is no comparison with the reported literature.
4. what precautions will take in order to store the sample or to remove the sample from one measurement to another measurement.
Reviewer 2 Report
This is a well-written paper containing interesting results which merit publication. In this paper, a high tensile conductive film: CMC-PEDOT:PSS was prepared based on CMC. This flexible film has little change in relative resistance under mechanical deformation, which has excellent mechanical properties and a good application prospect in the field of flexible sensor. For the benefit of the reader, a few minor revisions are list below:
1. When studying the effect of the ratio of CMC to PEDOT and DMSO on the electrical conductivity of CMC-PEDOT:PSS films(Figure 2a, Figure 2b), only the performance below the optimal ratio is presented with a poor property, but the performance data above the optimal ratio is not presented with a described property.
2. In the study, different formulations have been used to form films by drying method. The films may be formed with different thicknesses. And the different thicknesses may affect the corresponding performance of films. Please explain it in the text.
Reviewer 3 Report
Dear authors,
thank you very much for your investigations that is addressing an actual topic.
In your paper you work on the resistivity of stretchable films that are casted from different solutions/dispersions. The resulting layers are used for investigation of resistivity changes enabling the use at human skin for quite some different kinds of applications.
The article is well prepared. Please allow me some remarks and suggestions:
- (line 35) You might increase the interest for your article by adding a value of conductivity in the abstract
- (lines 66-81) This paragraph is very similar to the abstract. A first idea would be to omit this paragraph at all because all information can also be found in the abstract. Maybe you find a solution in shortening this paragraph while adding more content to it?
- (lines 90, 99) I personally don't like a line break between value and unit. Maybe you can use a 'secure space' that prevent a line break?
- (line 124) You give 0 % and 100 % of strain. I cannot find the dimensions in mm in either case?
- (line 127) In Fig 1b and 1c a scale bars are missing.
- (line 140) "1.5" should read "1:5"
- (line 143) a space is missing in "1:13,1:9"
- (line 161) Figure 2 / Figure S2: In my expectation a dependence between ratio PEDOT:PSS or DMSO and sheet resistance (ratio) should exist. Your measurements do not show this. It seems to have neglectable influence. Please discuss this situation and observation.
- (line 161) Fig 2f: there is some overlay by "Strain 50%"?
- (line 184) Fig 3a: please add the scale bar to the other three images
- (line 220) Fig 4. There is a mismatch of R/Ro and 0. R/Ro is 1 for R = Ro. Therefore, R/Ro (%) must be 100 %. So you'll need to rework the y-axis for every of the 9 graphs.
- (line 220) Fig 4 a-f: The required power is a very helpful parameter. What I miss is any description of circuitry and voltages / currents. Especially for application on the human skin, a voltage is a restricted parameter. Therefore, the information about how you have determined the given microwatts is missing.
- According to The International System of Units (SI) (5.4.7 Stating quantity values being pure numbers) the following statement is given: "The internationally recognized symbol % (percent) may be used with the SI. When it is used, a space separates the number and the symbol %." I'm aware that a discussion is ongoing whether using a space or not. My suggestion is to use a space according to the SI recommendation.
- You are using 20 reference sources. Among these are 4 sources (7, 9, 13, 19) where there is a significant overlap in authors to your current article. All these references are given together with other articles supporting the same statement. In my opinion, you might omit these 4 sources to follow a good scientific practise. If you think that you will miss any significant reference when omitting these 4 sources, please explain the additional benefit for each of the 4 sources that you intent to keep as reference.
